# Deep Learning-Based Speech Enhancement of an Extrinsic Fabry–Perot Interferometric Fiber Acoustic Sensor System

**DOI:** 10.3390/s23073574

**Published:** 2023-03-29

**Authors:** Shiyi Chai, Can Guo, Chenggang Guan, Li Fang

**Affiliations:** 1School of Science, Hubei University of Technology, Wuhan 430068, China; 2Hubei Engineering Technology Research Center of Energy Photoelectric Device and System, Hubei University of Technology, Wuhan 430068, China

**Keywords:** optical fiber sensor, external Fabry–Perot interferometer, speech enhancement, CV-CNN, CV-LSTM

## Abstract

To achieve high-quality voice communication technology without noise interference in flammable, explosive and strong electromagnetic environments, the speech enhancement technology of a fiber-optic external Fabry–Perot interferometric (EFPI) acoustic sensor based on deep learning is studied in this paper. The combination of a complex-valued convolutional neural network and a long short-term memory (CV-CNN-LSTM) model is proposed for speech enhancement in the EFPI acoustic sensing system. Moreover, the 3 × 3 coupler algorithm is used to demodulate voice signals. Then, the short-time Fourier transform (STFT) spectrogram features of voice signals are divided into a training set and a test set. The training set is input into the established CV-CNN-LSTM model for model training, and the test set is input into the trained model for testing. The experimental findings reveal that the proposed CV-CNN-LSTM model demonstrates exceptional speech enhancement performance, boasting an average Perceptual Evaluation of Speech Quality (PESQ) score of 3.148. In comparison to the CV-CNN and CV-LSTM models, this innovative model achieves a remarkable PESQ score improvement of 9.7% and 11.4%, respectively. Furthermore, the average Short-Time Objective Intelligibility (STOI) score witnesses significant enhancements of 4.04 and 2.83 when contrasted with the CV-CNN and CV-LSTM models, respectively.

## 1. Introduction

It is necessary to achieve high-quality voice communication in environments that have high temperatures, high pressure, strong radiation and strong electromagnetic effects. These environments create difficulties for conventional electroacoustic sensors and cause them not to work properly [1]. External Fabry–Perot interferometric (EFPI) acoustic sensors are widely used in special fields such as national defense and security [2], marine acoustic monitoring and positioning [3] and fuel pipeline leakage and positioning [4] because of their passive detection end, anti-electromagnetic interference, low loss, corrosion resistance and long-distance capabilities [5,6,7]. However, since the noise present in these environments can significantly degrade the perceptual quality and clarity of voice communication, voice enhancement is a much-needed task.

Speech enhancement is one of the most important and challenging tasks in speech applications, in which the goal is to suppress and reduce noise interference to extract useful speech signals in noisy backgrounds [8,9]. With the successful application of deep learning in the field of images [10,11,12,13,14], many scholars have begun to apply deep learning technology to speech enhancement. The existing speech enhancement methods can be divided into two categories: machine learning and deep learning. Regarding machine learning, early algorithms were generally implemented on shallow models and small datasets due to the limitations of computer hardware. Kim et al. developed a Gaussian mixture model (GMM)-based method for time–frequency (T-F) units according to the frequency band characteristics of input signals to determine the probability of speech and noise. However, this method models each frequency band separately, ignoring the correlation between frequency bands [15]. Han et al. used the support vector machine (SVM) method to identify speech-dominated and noise-dominated T-F units. Compared with the GMM, the SVM shows a better generalization ability, but it still loses some target speech when the noise speech energy is too high [16]. Chung et al. designed a training and compensation algorithm of class-conditioned basis vectors in a nonnegative matrix factorization (NMF) model for single-channel speech enhancement. The NMF algorithm is trained separately on clean speech and noise to reduce residual noise components that have similar characteristics to clean speech. However, when encountering speech or noise that does not appear in training, the performance of the algorithm will drop [17].

Regarding deep learning, neural networks can enhance noisy speech in the time–frequency (TF) domain or directly in the time domain. In the time-domain method, the neural network directly learns the mapping relationship of the time-domain waveform level, and the processing flow is relatively simple. A one-dimensional convolutional neural network is usually used. Between feature extraction and waveform restoration, a neural network with a temporal modeling ability, such as a recurrent neural network (RNN) [18] and a temporal convolutional network (TCN) [19], is used to enhance the effect of speech enhancement. However, a shared limitation of RNNs and TCNs is their inability to effectively capture long-term dependencies in speech signals that extend across multiple time frames. On the other hand, TF domain methods, which are also popular, perform speech enhancement by learning the masking relationship from the spectrogram of noises to the spectrogram of clean speech. It is believed that speech and noise are more separated after passing through the short-time Fourier transform (STFT), which usually uses a convolutional encoder–decoder (CED) or U-Net framework [20]. After the STFT was performed, TF domain methods can take the complex-valued spectrogram as input and then decompose the magnitude and phase into real and imaginary parts in Cartesian coordinates. These methods solve the long-standing problem of the phase being difficult to estimate. Recently, Choi et al. proposed deep complex U-Net (DCUnet), which utilizes deep complex-valued convolutional layers for the near-perfect enhancement of speech [21]. Cao et al. proposed a generative adversarial network to model temporal and frequency correlations and achieved extremely high performance [22]. Park et al. proposed a multi-view attention network to improve the accuracy of feature extraction [23].

The EFPI acoustic sensor is a special microphone that converts sound waves into an optical signal. The speech signal needs to be received by the EFPI acoustic sensor and demodulated by the demodulation system to restore the sound signal. Unlike conventional microphones, some optical parameters (such as the light intensity, phase and polarization state) will also affect the acoustic characteristics of the EFPI acoustic sensor and cause frequency responses and noise [24,25]. Although deep learning has achieved extremely high performance in speech enhancement, most speech enhancement models are designed for conventional microphones and may not work perfectly on EFPI acoustic sensors.

In this paper, we propose a hybrid deep learning architecture that combines a complex-valued convolutional neural network and a long short-term memory (CV-CNN-LSTM) for speech enhancement of fiber-optic EFPI acoustic sensors. The CV-CNN and LSTM are set as the primary framework for the neural network, and the TF domain features of the signal are extracted through the STFT, completing the purpose of speech enhancement. Compared with a simple CV-CNN and LSTM, CV-CNN-LSTM is the best with regard to the speech enhancement performance test.

## 2. Basic Configuration and Data Acquire

### 2.1. Basic Configuration

Our EFPI acoustic sensor system with three-wavelength demodulation is schematically shown in Figure 1. An Er-doped amplified spontaneous emission (ASE) broadband source with an output power of 100 mw was used in our experiment. The output light from the ASE source is incident to the EFPI sensor head through an optical circulator. The EFPI sensor head is formed by the cleaved end face of a single-mode-fiber (SMF) and silicon nitride diaphragm, which creates two reflective mirrors in the EFPI cavity. The cavity length of our EFPI sensor is approximately 100 µm. When an acoustic wave is applied to the silicon nitride diaphragm, it vibrates with the applied sound pressure, which modulates the cavity length of the EFPI and consequently induces the phase change of the interferential output light. The reflected light beams modulated with the phase signal are incident to a wavelength division multiplexer (WDM) through the optical circulator and divided into three beams according to their wavelengths. The wavelength interval of the three wavelengths is chosen according to the free spectrum range (FSR) of our EFPI sensor head. Each beam is then collected by a photodetector (PD) and converted into a voltage signal. The voltage signals are collected by data acquisition (DAQ) and processed by a computer. A loudspeaker excited by an audio analyzer is used as the acoustic source, which can generate sinusoidal acoustic waves at a specific frequency.

It is constructed entirely within a silent room to guarantee the system remains unaffected by external conditions during testing. This provides an environment of utmost silence to eliminate or minimize the impact of external noise and echoes on the recorded or tested audio.

### 2.2. Demodulation Principle

The intensity of the reflected interferential light at the three quadrature wavelengths can be expressed as
(1)$$\begin{array}{c} I_{i}=A+Bcos\left(\varphi_{i}\right)\\ \varphi_{i}=\frac{4n\pi }{\lambda_{i}}d_{t} \end{array},(i=1,\;2,\;3)$$
where $\lambda_{i}$ is the output wavelength, A is the DC component of the interferometric fringe, *B* is the interferometric fringe visibility, n is the refractive index of the EFPI cavity, n = 1 and $d_{t}$ is the cavity length modulated by the vibration signal. The relationship of the three wavelengths is calculated as follows to satisfy the quadrature relationship between the three output wavelengths.
(2)$$\begin{array}{c} 4\pi L/\lambda_{1}+2\pi /3=4\pi L/\lambda_{2}\;\\ 4\pi L/\lambda_{1}+4\pi /3=4\pi L/\lambda_{3} \end{array}$$
where L  is the initial cavity length.

Considering that the three wavelengths cannot be completely equal in power and the phase difference cannot strictly meet 2π/3, the three interferometric signals received by the DAQ can be described by
(3)$$\left\{\begin{array}{c} f_{1}=D_{1}+E_{1}\cos\left[\phi \left(t\right)+2\pi /3+\varphi_{1}\right]\\ f_{2}=D_{2}+E_{2}\cos\left[\phi \left(t\right)+\varphi_{2}\right]\\ f_{3}=D_{3}+E_{3}\cos\left[\phi \left(t\right)-2\pi /3-\varphi_{3}\right]\; \end{array}\right.$$
where $D_{1}$, $D_{2}$ and $D_{3}$ are the DC components of the interferometric fringe. $E_{1}$, $E_{2}$ and $E_{3}$ are the fringe visibility. $\varphi_{1}$, $\varphi_{2}$ and $\varphi_{3}$ are the phase deviations of the three outputs. $\phi \left(t\right)$ is the external disturbance signal, and t represents the time. A new output can be obtained by taking the average value of the two signals $f_{1}$ and $f_{2}$ output in the asymmetric state, and the new output $p_{1}$ can be expressed as
(4)$$p_{1}=\left(D_{1}+D_{2}\right)/2+E_{4}\cos\left[\phi \left(t\right)+2\pi /3+\varphi_{4}\right]$$
where $E_{4}$ is the interference fringe visibility of the new signal $p_{1}$, $\varphi_{4}$ is the phase deviation of $p_{1}$ and the DC component and phase deviation are the most critical factors affecting the 3 × 3 coupler algorithm [26]. Similarly, by performing the same operation on $f_{2}$ and $f_{3}$, another new signal, P2 can be expressed as
(5)$$p_{2}=\left(D_{2}+D_{3}\right)/2+E_{5}\cos\left[\phi \left(t\right)+2\pi /3+\varphi_{5}\right]$$
where $E_{5}$ is the DC component of $p_{2}$, $\varphi_{5}$ is the phase deviation of $p_{2}$. Moreover, $p_{1}$, $p_{2}$ and *f*_2_ are used as new inputs to the 33 coupler algorithm. It can be seen from the three new signals that the difference between the DC components is very small after the operation. Compared with the original signals, the DC components of the three new signals are closer in value. In addition, the phase deviation is relatively small, and it is compressed to be smaller after the operation. Thus, the errors caused by the obtained $\varphi_{4}$ and $\varphi_{5}$ can be ignored.

### 2.3. Feature Extraction

The short-time Fourier transform (STFT) is used to extract the time–frequency domain features of speech signals. The STFT has excellent time–frequency resolution, which means that it can accurately localize changes in the frequency content of a signal over time. This makes it a powerful tool for analyzing time-varying signals. For a discrete signal $x\left(n\right)$ of length N, the discrete STFT at the frequency f and the short interval at the tth moment can be expressed as follows
(6)$$X\left(f,t\right)=\sum_{n=0}^{N-1}x\left(n\right)w\left(n-tB\right)e^{-j2\pi fn}$$
where $w\left(n\right)$ is the window function and B is the hop length. The spectrogram representation of speech data is influenced by both the window size and the hop length. Specifically, the window size primarily affects the frequency resolution, while the time resolution is primarily influenced by the hop length. Figure 2 shows the spectrograms of a clean speech signal and a noisy speech signal.

## 3. Neural Networks

### 3.1. Complex-Valued Convolution

Complex-valued convolution is an operation in mathematics and signal processing in which two complex-valued functions are convolved with each other. A complex-valued spectrogram obtained by the STFT of a speech signal can be decomposed into real and imaginary parts in Cartesian coordinates. The CV-CNN convolves the real part and imaginary part, respectively, with two complex-valued functions. The complex-valued convolution filter, also known as kernels, is defined as W=A+iB, where both A and B are real-valued matrices. The input complex matrix X is defined as $X=X_{r}+iX_{i}$, and the complex-valued convolution operation on W with X is done by $W*X=\left(A*X_{r}-B*X_{i}\right)+i\left(B*X_{r}+A*X_{i}\right)$. The operation process of complex-valued convolution is shown in Figure 2.

The complex-valued convolutional layers constitute the main structural elements of a CV-CNN and extract features from the complex-valued input data. A complex-valued convolution can be implemented as a set of four convolutions with real-valued functions. The two complex-valued functions are separated into their real and imaginary components, and each component is convolved separately with the corresponding component of the other function. Thus four real-valued convolutions can be combined to form the final complex-valued convolution. These convolution kernels are two-dimensional (2D) since the one-dimensional (1D) speech time waveforms have been transformed into a complex-valued spectrogram with the application of the STFT. Since the spectrogram is complex-valued, the dot product is computed separately for the real and imaginary parts of the filter and the spectrogram, resulting in a complex-valued output at each position.

Multiple hidden layers are included in a typical complex-valued convolutional structure, which can extract features of high-dimensional data and learn nonlinear relationships adaptively. We add a batch normalization (BN) layer and a Leaky ReLu (LR) layer after each convolutional layer. The BN layer can improve the training stability and convergence speed of the network by normalizing the activations of the previous layer. When used after each convolutional layer, the BN layers can help to stabilize the training process by maintaining the variance and mean of the activations close to zero and one, respectively. The LR activation functions are a variant of the ReLu that allows a small gradient when the input is negative. The LR layer can help to introduce nonlinearity and improve the ability of the network to learn complex representations of the input data. A complete complex-valued convolutional network structure is shown in Figure 3.

### 3.2. Long Short-Term Memory

When deep learning is applied to sequential data such as music, video and speech, it is important to model the long-term dependencies in the time series. RNNs are designed to process sequential data using the mechanism of recurrent feedback. However, RNNs are notoriously difficult to train due to the vanishing and exploding gradient issues. To overcome these problems, the LSTM was proposed as a special type of RNN. There are three important components inside the LSTM, including the input gate, forgetting gates and output gates. These gates allow the network to selectively retain or discard information from previous time steps, ensuring that relevant features of the sequential data are accurately captured. By selectively controlling the flow of information through the network, LSTMs can effectively model long-term dependencies and have become a widely used approach in various applications of sequential data analysis.

CV-CNN excels at learning time–frequency domain features in speech signals, but its ability to capture time dependence and long-term context information is limited. In contrast, LSTMs are adept at capturing long-term dependencies in speech signals, which may span across multiple time frames. This makes them well-suited for modeling the temporal structure of speech and reducing the impact of noise interference. By integrating LSTM with CV-CNN, we can build a model that effectively handles speech signals of different lengths without requiring extensive preprocessing. In order to be able to handle complex-valued features extracted by the CV-CNN, we use the CV-LSTM in the proposed model. Similar to CV-CNN, considering the real and imaginary components of the complex input $X_{r}$ and $X_{i}$, the output of the CV-LSTM, $F_{out}$, can be defined as
(7)$$\left\{\begin{array}{c} F_{rr}={\mathrm{LSTM}}_{r}\left(X_{r}\right);F_{ir}={\mathrm{LSTM}}_{r}\left(X_{i}\right)\\ F_{ri}={\mathrm{LSTM}}_{i}\left(X_{r}\right);F_{ii}={\mathrm{LSTM}}_{i}\left(X_{i}\right)\\ F_{out}=\left(F_{rr}-F_{ii}\right)+j\left(F_{ri}+F_{ir}\right) \end{array}\right.$$
where ${\mathrm{LSTM}}_{r}$ and ${\mathrm{LSTM}}_{i}$ represent two traditional LSTMs of the real part and the imaginary part.

### 3.3. Target and Loss Function

The target of model training is a complex ratio mask (CRM). CRM is a technique used in speech enhancement. It is a complex-valued function that estimates the ratio between the desired speech signal and the interfering noise signal. The CRM is computed by taking the complex ratio of the time–frequency representations of the desired speech and the noisy signal. The resulting CRM is then applied to the noisy signal to suppress the noise and enhance the desired speech signal. The CRM is able to capture the phase information of the signal, which is useful when the noise around the EFPI acoustic sensor is unstable, and its phase changes over time; CRM can be defined as
(8)$$CRM=\frac{Y_{r}S_{r}+Y_{i}S_{i}}{Y_{r}^{2}+Y_{i}^{2}}+j\frac{Y_{r}S_{i}-Y_{i}S_{r}}{Y_{r}^{2}+Y_{i}^{2}}\;$$
wre $Y_{r}$ and $Y_{i}$ denote the real and imaginary parts of the noisy complex-valued spectrogram obtained by the STFT, respectively. The real and imaginary parts of the clean speech complex-valued spectrogram obtained by the STFT are represented by $S_{r}$ and $S_{i}$.

The loss function measures the discrepancy between the predicted output of the model and the true output, and the goal of the model is to minimize this discrepancy. In the present study, the scale-invariant signal-to-noise ratio (SI-SNR) is utilized. The SI-SNR is a metric commonly used to evaluate the performance of speech separation or source separation algorithms. It measures the ratio of the energy of the target speech signal to the energy of the interference or noise signal while being insensitive to the amplitude scaling of the separated signals. The SI-SNR can be defined as
(9)$$\mathrm{SI}-\mathrm{SNR}=10{*\log}_{10}(\frac{{\left|\left|s\right|\right|}^{2}}{{\left|\left|s-y\right|\right|}^{2}})$$
where s is the reference signal and y is the estimated signal; the ||.|| operator denotes the L2 norm.

### 3.4. The CV-CNN-LSTM Model

In the present study, the CV-CNN-LSTM model mainly adopts the encoder–decoder structure. The encoder receives input and compresses it into a reduced representation, which is subsequently forwarded to a decoder. The decoder then generates an output based on this compressed representation. The encoder–decoder framework is commonly used in applications such as natural language processing, speech and audio processing and image and video processing.

In the CV-CNN-LSTM model, the encoder is the complete complex-valued convolution network structure mentioned in Section 3.1. This encoder is composed of a complex-valued convolution layer, batch normalization layer and LeakyReLu layer. The decoder is similar in structure to the encoder, except that all 2D convolution functions are replaced by 2D transposed functions. The structure of the CV-CNN-LSTM model is shown in Figure 4. This model consists of six encoder blocks, six decoder blocks and one CV-LSTM) layer. Moreover, the FC is a fully connected layer. The fully connected layers in the model are used to learn nonlinear combinations of features at a higher level of abstraction.

To improve the performance and training efficiency of the encoder–decoder architecture, we use skip connections between the encoder and decoder. Skip connections enable the network to reuse learned features from earlier layers in later layers, which can help to preserve important information and prevent that information from being lost during training. Additionally, with skip connections, the network can converge faster because it is easier to learn identity maps than to learn complex maps from scratch.

## 4. Results and Discussion

To ensure the normal demodulation of the optimized 3 × 3 algorithm, we first conducted a performance test in an anechoic chamber in which the three wavelengths are selected according to Equation (2): 1546.92 nm ($\lambda_{1}$), 1550.92 nm ($\lambda_{2}$) and 1554.94 nm ($\lambda_{3}$). The NI-USB6210 is used to collect and process the signals of the three PDs. The demodulated signals are connected to the audio analyzer through the earphones for testing. The demodulation effects of the different frequency signals are shown in Figure 5. To obtain more detailed indicators, we used the audio analyzer test to evaluate the signal-to-noise ratio. The results, which reach a signal-to-noise ratio of 62 DB, are shown in Figure 6.

In this study, our dataset is constructed from the 28 speakers dataset (28spk) [27], which contains speech clips from 28 people. We randomly selected 6000 utterances from the 28spk corpus and divided them into three parts. There were 4800 utterances in the training set, 660 utterances in the validation set and 540 utterances in the test set. All utterances were played in the anechoic chamber through a high-quality speaker, and EFPI acoustic sensors were used to capture the speech signals. In the end, the audio was clipped to ten seconds, and we obtained approximately 16 h of paired clean and noisy utterances. The evaluation set is generated by randomly selecting utterances from the speech set and the noise set and mixing them at three signal-to-noise ratios (SNRs) (5 DB, 10 DB and 15 DB). According to the frequency range of the voice call, all the speech signals were sampled at 16 kHz. The design and implementation of the deep learning model for the speech enhancement of the EFPI acoustic sensor were held in a Python 3.8.3 environment using the deep learning tool Torch. Pytorch 1.7 was used as the backend of the Torch library. All experiments were performed on a desktop computer featuring an Intel Core i7-10700 2.90 GHz CPU, 32 GB RAM memory and a 10 GB NVIDIA GeForce RTX 3080 GPU.

Regarding the CV-CNN-LSTM model, the complex-valued spectrogram was extracted by using a Hamming window. The window length and hop size are 25 ms and 6.25 ms, respectively, and the FFT length is 512. The number of channels for the CV-CNN-LSTM is {32, 64, 128, 256, 256, 256}. The kernel size and stride are set to (5, 2) and (2, 1), respectively. The CV-LSTM layer uses a two-layer structure, the parameters of the two layers are the same, and each layer contains 128 hidden units. We chose Adam as the optimizer, set the initial learning rate to 0.001 and used ExponentialLR to control the change in the learning rate. We compare several models, including CNN, LSTM, CV-CNN and CV-LSTM, on the same dataset. The CNN model is structured with six 2D convolutional layers, each accompanied by a batch normalization layer and succeeded by a max-pooling layer. Channel counts for the CNN model are specified as {16, 32, 64, 64, 128, 128}, while kernel sizes and strides are consistently set to (3, 3) and (1, 1). The LSTM model contains two LSTM layers; each layer has 256 units, and the output layer is a fully-connected layer. Comprising six CV-CNN layers, the CV-CNN model boasts channel numbers of {16, 32, 64, 64, 128, 128} and identical kernel size and stride settings as its CNN counterpart. Lastly, the CV-LSTM model is composed of two CV-LSTM layers, and each LSTM layer has 256 units and utilizes two separate fully-connected layers to deliver the real and imaginary components of the results, respectively.

The evaluation of speech enhancement model performance is conducted using two widely-accepted and complementary metrics: Perceptual Evaluation of Speech Quality (PESQ) [28] and Short-Time Objective Intelligibility (STOI) [29]. PESQ scores generally range from −0.5 to 4.5, with higher values indicating superior speech quality. The STOI metric provides a normalized score between 0 and 1, where higher values correspond to greater speech intelligibility. Table 1 and Table 2 show the comparison of PESQ and STOI scores between the proposed model and other models. It can be seen that the proposed model has the best test results on data with different SNRs.

Furthermore, additional experiments were conducted to optimize the proposed model. In particular, the performance of the proposed model was evaluated while changing some of its parameters, namely, the window length of the STFT and the number of complex-valued convolutional layers. The window length is a key parameter in the STFT for the feature extraction of speech signals. Using a window length that is too long can result in poor time resolution. This results in a loss of important spectral information and increased spectral leakage. Conversely, if the window length is too short, it will result in a poor frequency resolution. This can lead to poor separation of the speech and noise components in the frequency domain, resulting in a low quality of enhanced speech. The experimental results of the window length comparison are shown in Table 3 and Table 4. It can be seen that the optimal window length is approximately 25 ms. The enhancement effect of the CV-CNN-LSTM model that uses different window lengths is shown in Figure 7. The noise signal used in Figure 6 is a speech recording at an SNR of 10 dB.

The number of complex-valued convolutional layers also plays a significant role in the model’s performance. If the number is too low, the model may fail to capture crucial input data features effectively. On the other hand, excessive layers may lead to overfitting, resulting in good training set scores but poor performance on the test set. Table 5 and Table 6 show the performance of the proposed model for different numbers of complex-valued convolutional layers. The model was trained and tested using four to eight complex-valued convolutional layers in a structure similar to the one displayed in Figure 4. As can be observed, both PESQ and STOI scores are highest when using six complex-valued convolutional layers in the CV-CNN-LSTM model.

## 5. Conclusions

In this paper, speech enhancement techniques based on fiber-optic EFPI acoustic sensors are studied. First, the speaker’s speech signal is demodulated by the fiber-optic EFPI acoustic sensor demodulated based on the 3 × 3 coupling algorithm, and then the speech signal is edited, and the edited speech signal is subjected to STFT to extract spectral features. The overall structure of the CV-CNN-LSTM model is implemented by combining CV-CNN and CV-LSTM. Among them, CV-CNN is suitable for processing complex-valued spectrogram data, while CV-LSTM is good at capturing the characteristics of sequential data related to time series. Experimental results show that the CV-CNN-LSTM model can achieve better performance than other models in terms of PESQ score and STOI score.

The speech enhancement technology in this paper is expected to be applied to the fields where traditional methods cannot be applied, such as high magnetic field environments, flammable and explosive environments and high electric field environments. Of course, the fiber-optic EFPI acoustic sensor process in this paper is more complex than electrical-based acoustic sensors. With the reduction of the cost of optoelectronic devices, the technology is expected to be used in fields such as deserts and polar regions.

## Figures and Tables

**Figure 1 sensors-23-03574-f001:**
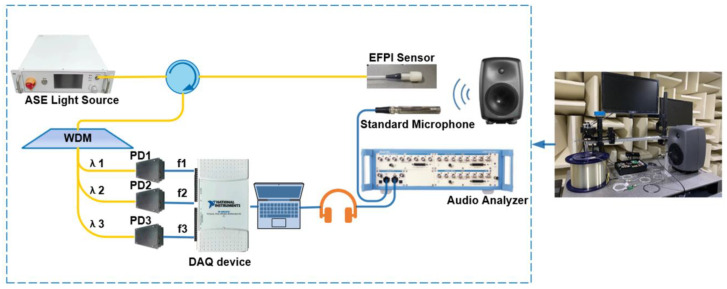
Schematic of the three-wavelength EFPI acoustic sensor system.

**Figure 2 sensors-23-03574-f002:**
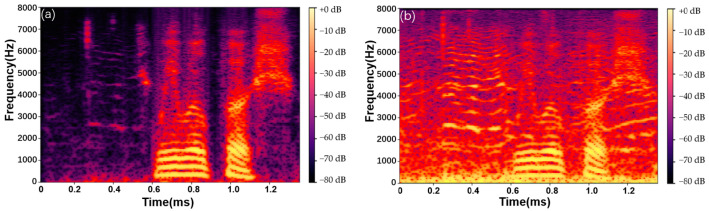
The spectrogram of (**a**) a clean speech signal and (**b**) a noisy speech signal.

**Figure 3 sensors-23-03574-f003:**
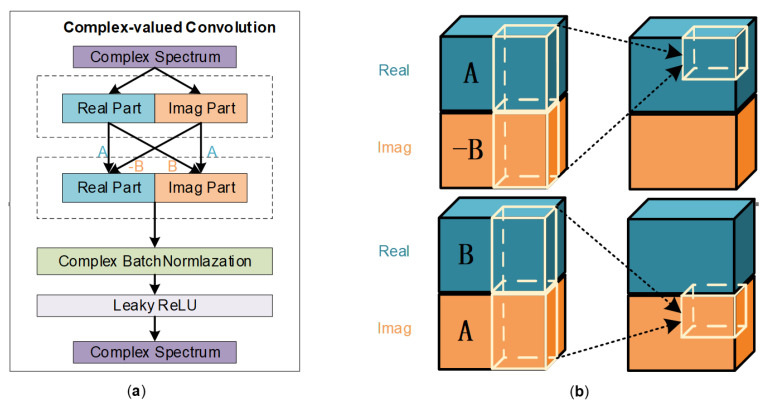
(**a**) The specific structure of complex-valued convolution. (**b**) Calculation of real and imaginary parts in complex-valued convolution operations.

**Figure 4 sensors-23-03574-f004:**
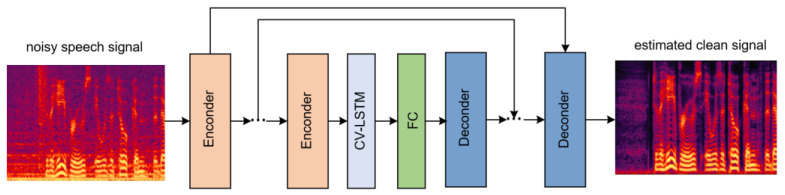
The structure of CV-CNN-LSTM model for speech enhancement.

**Figure 5 sensors-23-03574-f005:**
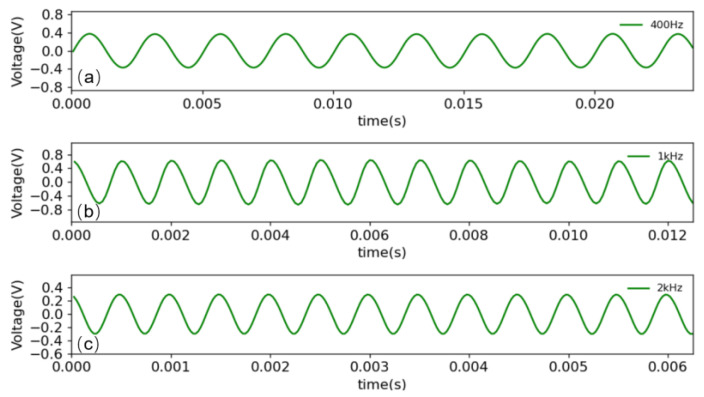
Demodulation results at different frequencies: (**a**) the demodulated signal of 400 Hz, (**b**) the demodulated signal of 1 KHz, (**c**) the demodulated signal of 2 KHz.

**Figure 6 sensors-23-03574-f006:**
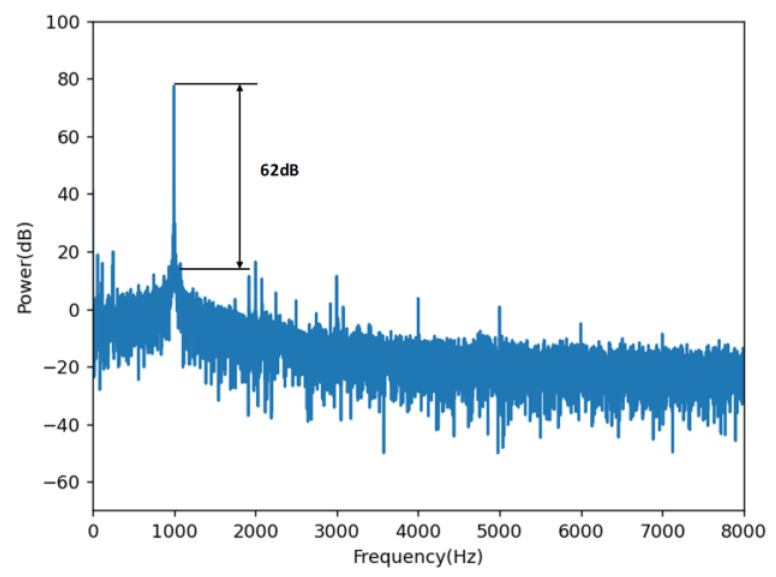
The result of the signal-to-noise ratio.

**Figure 7 sensors-23-03574-f007:**
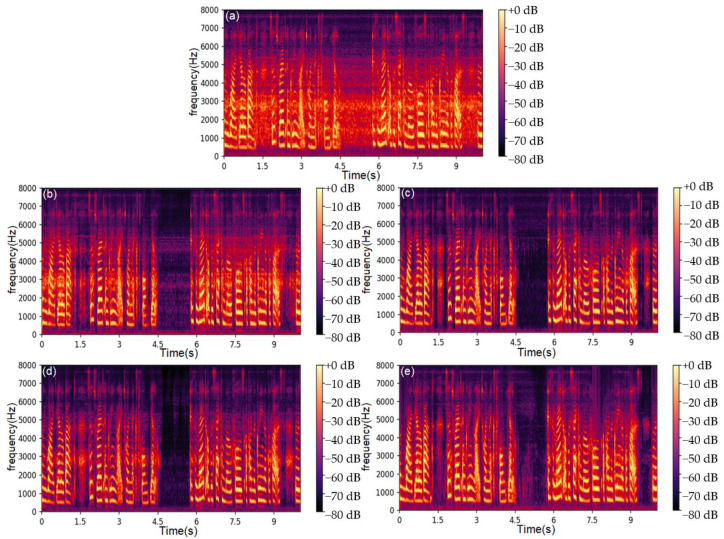
Test results of models with different window lengths: (**a**) STFT spectrogram of original speech signal, (**b**) model with a window length of 15 ms, (**c**) model with a window length of 25 ms, (**d**) model with a window length of 40 ms, (**e**) model with a window length of 64 ms.

**Table 1 sensors-23-03574-t001:** The PESQ scores of different models at different SNRs.

Model	5 dB	10 dB	15 dB	Ave.
CNN	2.347	2.544	2.627	2.506
CV-CNN	2.724	2.835	3.045	2.868
LSTM	2.701	2.846	2.928	2.825
CV-LSTM	2.746	2.897	3.107	2.917
CV-CNN-LSTM	2.948	3.135	3.361	3.148

**Table 2 sensors-23-03574-t002:** The STOI (in %) scores of different models at different SNRs.

Model	5 dB	10 dB	15 dB	Ave.
CNN	77.97	83.15	85.08	82.07
CV-CNN	86.73	87.65	91.97	88.78
LSTM	86.59	87.71	89.35	87.88
CV-LSTM	86.94	88.61	92.42	89.99
CV-CNN-LSTM	89.51	93.14	95.83	92.82

**Table 3 sensors-23-03574-t003:** The PESQ scores for models with different window lengths.

Win_Length (ms)	5 dB	10 dB	15 dB	Ave.
64	2.619	2.826	3.049	2.831
40	2.801	2.904	3.212	2.972
25	2.948	3.135	3.361	3.148
15	2.745	2.843	3.108	2.899

**Table 4 sensors-23-03574-t004:** The STOI (in %) scores for models with different window lengths.

Win_Length (ms)	5 dB	10 dB	15 dB	Ave.
64	84.91	87.52	91.98	88.14
40	87.29	88.76	94.51	90.19
25	89.51	93.14	95.83	92.82
15	86.93	87.61	92.38	88.97

**Table 5 sensors-23-03574-t005:** The PESQ scores for models with different complex-valued convolutional layers.

No. of Complex-Valued Convolutional Layers	5 dB	10 dB	15 dB	Ave.
4	2.768	2.921	3.126	2.938
5	2.813	3.019	3.205	3.012
6	2.948	3.135	3.361	3.148
7	2.804	2.977	3.156	2.979
8	2.932	3.044	3.275	3.083

**Table 6 sensors-23-03574-t006:** The STOI (in %) scores for models with different complex-valued convolutional layers.

No. of Complex-Valued Convolutional Layers	5 dB	10 dB	15 dB	Ave.
4	87.34	89.27	93.05	92.87
5	87.28	91.63	94.22	91.04
6	89.51	93.14	95.83	92.82
7	87.16	91.37	93.49	90.67
8	89.42	91.28	94.55	91.75

## Data Availability

Not applicable.

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
