# Peer review of "Deep Learning-Based Speech Enhancement of an Extrinsic Fabry–Perot Interferometric Fiber Acoustic Sensor System"

_sensors, 2023, doi:10.3390/s23073574_

Round 1

Reviewer 1 Report

To achieve high-quality voice communication technology without noise interference in flammable, explosive and strong electromagnetic environments, the speech enhancement technology of a fiber-optic external Fabry‒Perot interferometric (EFPI) acoustic sensor based on deep learning is studied in this paper. This paper proposed a complex-valued convolutional neural network and a long short-term memory (CV-CNN-LSTM) model for speech enhancement in the EFPI system.

The experimental results show that the CV-CNN-LSTM model proposed in this paper has good speech enhancement performance, and the average perceptual evaluation of speech quality (PESQ) score tested is 3.148. Compared with the CVCNN model and the LSTM model, the PESQ score of this model is 9.7% and 11.4% 20 higher, respectively.

The proposed method sounds good, but need to add some experiment results to reach a valid conclusion.

Some issues are as follow:

1. For ablation studies, table 1 should include all these experiment result: CNN only, LSTM only, CV-CNN, CV-LSTM and CV-CNN-LSTM. Then can get valid conclusion.

2.Usually for speech enhancement evaluation, 2 objective measurements should be considered. PESQ (Perceptual Evaluation of Speech Quality) is only one of them, to evaluate speech quality. The other STOI (Short Time Objective Intelligibility) is also needed to evaluate the intelligibility. Why intelligibility evaluation part is missing?

Author Response

We would like to thank you for your careful reading, helpful comments, and constructive suggestions, which has significantly improved the presentation of our manuscript.

Reviewer 2 Report

interferometric fiber acoustic sensor system. Overall, the structure of this paper is well organized, and the presentation is relatively clear. The idea is interesting and potential. However, there are still some crucial problems that need to be carefully addressed before a possible publication. More specifically,

1.     The motivations or remaining challenges are not so clear or what kinds of issues or difficulties are this task that is facing. Please give more details and discussion about the key problems solved in this paper, which is largely different from existing works.

2.     A deep literature reviews should be given, particularly advanced and SOTA deep learning or AI models. Therefore, the reviewer suggests discussing some related works by analyzing the following papers in the revised manuscript, e.g., 10.1109/TGRS.2020.3016820, 10.1109/TGRS.2020.3015157, 10.1109/TIP.2022.3228497.

3.     Please clarify the contributions to this field, for example, which are the existing ones and which are your own ones?

4.     The reviewer is wondering how about the computational complexity of the proposed method?

5.     What are the differences in techniques between the proposed method and existing methods?

6.     It is well-known that the data usually tend to suffer from various degradation, noise effects, or variabilities in the process of imaging. Please give the discussion and analysis by referring to the paper titled by e.g., An Augmented Linear Mixing Model to Address Spectral Variability for hyperspectral unmixing. The reviewer is wondering what will happen if the proposed method meets the various variabilities.

7.     Some future directions should be pointed out in the conclusion.

Author Response

(The authors gave the same response as above.)

Round 2

Reviewer 1 Report

authors have modified the paper, and this version looks good to me.

Reviewer 2 Report

The authors have well addressed the reviewer's concerns. No more comments.